# EFFECTIVE RESISTANCE REWIRING: A SIMPLE TOPOLOGICAL CORRECTION FOR OVER-SQUASHING

**Bertran Miquel-Oliver**[1,2]**, Manel Gil-Sorribes**[3]**, Victor Guallar**[1,4,*]**, Alexis Molina**[3,*]

[1]Barcelona Supercomputing Center (BSC), Barcelona, Spain
[2]Universitat Politècnica de Catalunya - BarcelonaTech (UPC), Barcelona, Spain
[3]Nostrum Biodiscovery, Barcelona, 08029, Spain
[4]ICREA, Barcelona, Spain
*alexis.molina@nostrumbiodiscovery.com, victor.guallar@bsc.es

## ABSTRACT

Graph Neural Networks (GNNs) struggle to capture long-range dependencies due to over-squashing, where information from exponentially growing neighborhoods must pass through a small number of structural bottlenecks. While recent rewiring methods attempt to alleviate this limitation, many rely on local criteria such as curvature, which can overlook global connectivity bottlenecks that restrict information flow. We introduce Effective Resistance Rewiring (ERR), a simple topology correction strategy that uses effective resistance as a global signal to detect structural bottlenecks. ERR iteratively adds edges between node pairs with the largest resistance while removing edges with minimal resistance, strengthening weak communication pathways while controlling graph densification through a fixed edge budget. The procedure is parameter-free beyond the rewiring budget and relies on a single global measure aggregating all paths between node pairs. Beyond evaluating predictive performance on GCN model, we analyze how rewiring affects message propagation. By studying cosine similarity between node embeddings across layers, we study how the relationship between initial node features and learned representations evolves during message passing, comparing graphs with and without rewiring. his analysis helps determine whether performance gains arise from improved long-range communication. Experiments on homophilic (Cora, CiteSeer) and heterophilic (Cornell, Texas) graphs, including directed settings with DirGCN, reveal a fundamental trade-off between over-squashing and oversmoothing, losing representation diversity across layers. Resistance-guided rewiring improves connectivity and signal propagation but can accelerate representation mixing in deep models. Combining ERR with normalization techniques (e.g., PairNorm) stabilizes this trade-off and improves performance, particularly in heterophilic settings.

## 1 INTRODUCTION

Graph neural networks (GNNs) have become a standard approach for learning on relational data, based on message passing which provides a strong inductive bias: node representations are updated by aggregating information from neighbors, so predictions can exploit both features and connectivity (Kipf & Welling, 2017). Yet the same mechanism imposes constraints on how information can propagate across a graph, and these constraints arise even when optimization and expressivity are not the dominant bottlenecks (Alon & Yahav, 2020; Di Giovanni et al., 2023a). In contrast with other models such as Neural Networks or Transformers, message passing models face a tight balance with depth. With too few layers, information remains local and the model cannot exploit non-local evidence, hence losing generalization (Black et al., 2023). With too many layers, repeated aggregation drives representations toward collapse, reducing discriminability between different class nodes (Chen et al., 2020; Zhou et al., 2021; Rusch et al., 2023).

A classical explanation for depth degradation comes from viewing common propagation rules as smoothing operators on the graph. In particular, Graph Convolutional Networks (GCNs) can be interpreted as a form of Laplacian smoothing (Li et al., 2018). Moderate smoothing can denoise features and stabilize learning, but excessive smoothing makes node embeddings increasingly similar and eventually uninformative, a phenomenon referred to as oversmoothing (Rusch et al., 2023). This behavior has been quantified using topological and energy-based measures (Chen et al., 2020; Zhou et al., 2021), and analyzed theoretically as a trade-off between beneficial and harmful smoothing (Keriven, 2022). On the practical side, normalization techniques such as PairNorm aim to prevent representation collapse and enable deeper message passing without architectural changes (Zhao & Akoglu, 2019). At the opposite extreme, shallow models can fail to incorporate non-local signals, a limitation often discussed as underreaching (Black et al., 2023). However, depth alone does not explain why long-range dependencies remain difficult even when receptive fields are increased.

Over-squashing pinpoints a distinct topological obstruction to long-range reasoning in message passing GNNs. As depth increases, the receptive field of a node can grow rapidly, potentially encompassing many nodes. Nevertheless, information from this expanding neighborhood must traverse the graph connectivity and be repeatedly aggregated into fixed-size vectors. When many distant nodes can influence a target only through a narrow set of intermediate nodes or edges, message passing must "squeeze" a rapidly growing number of signals through a limited number of channels, causing distant information to have vanishing influence on the learned representation (Alon & Yahav, 2020). Crucially, recent evidence suggests that the topology of the input graph is the dominant driver of this effect. Architectural choices such as width or depth can modulate over-squashing, but bottlenecks induced by global connectivity patterns remain the primary limitation (Di Giovanni et al., 2023a). This bottleneck view therefore shifts the focus from architecture alone to the interaction between message passing and graph topology. As an example, two graphs with similar size can induce very different levels of over-squashing depending on where bottlenecks occur and how many alternative routes exist for information to flow.

**Geometry and topology perspectives on over-squashing.** Recent work has formalized over-squashing through geometric and topological descriptors of graphs. Curvature-based analyses connect bottlenecks to negatively curved edges via Jacobian sensitivity and motivate topology modifications that target local bottlenecks. (Topping et al., 2022). A complementary viewpoint emphasizes random-walk accessibility. In such scenarios, if information must traverse a narrow bottleneck, the expected time for a random walk to travel between two nodes (and return) can become large, indicating weak long-range influence (Di Giovanni et al., 2023b). At a high level, both lenses highlight that over-squashing is primarily driven by the graph topology rather than solely by architectural choices (Di Giovanni et al., 2023a). Nevertheless, curvature criteria are often local in nature, while random-walk quantities such as commute time can be informative but less direct as a design signal for topology interventions. This motivates seeking a global, pairwise measure that retains the random-walk interpretation while maintaining multi-path connectivity. Such measurements, however, come at a high computational expense.

**Effective resistance as a measure of over-squashing.** Effective resistance provides a global and interpretable measure of connectivity that is well suited to diagnosing over-squashing. An intuition comes from the electrical-network analogy, where a graph can be viewed as a network transporting information, and the resistance distance captures how easily two nodes can communicate through the available connections (Doyle & Snell, 1984; Klein & Randić, 1993). Because current in an electrical network flows through all routes in parallel, effective resistance aggregates the contribution of all paths between two nodes rather than focusing on shortest paths alone. It is also proportional to random-walk commute time, thereby retaining a natural accessibility interpretation while remaining explicitly multi-path (Black et al., 2023). Recent work has established a formal connection between effective resistance and over-squashing. In particular, Black et al. (2023) show that the influence of one node on another through message passing can be upper bounded by quantities related to their effective resistance, via bounds on suitable norms of the Jacobian of node embeddings with respect to the input features. Under this perspective, pairs of nodes with large effective resistance correspond to interactions whose influence is strongly attenuated during message passing, which is consistent with the notion of over-squashing. Their work further proposes reducing oversquashing by adding edges that minimize the total effective resistance of the graph. In this work, we adopt effective resistance primarily as a diagnostic signal for pairwise bottlenecks. Rather than globally

minimizing total resistance through edge addition, we focus on identifying node pairs whose high resistance indicates weak long-range connectivity. This perspective allows us to directly target the structural locations where information flow is most constrained. Reducing resistance between such distant nodes can strengthen end-to-end influence pathways and improve long-range communication when predictions depend on non-local interactions (Di Giovanni et al., 2023a; Black et al., 2023). Finally, effective resistance also admits connections to curvature-like notions on graphs, providing a bridge between global connectivity perspectives and geometric views of graph structure (Devriendt & Lambiotte, 2022).

**Rewiring must balance bottlenecks and smoothing.** Topology interventions, however, must respect the oversmoothing trade-off. Adding edges can create alternative routes and relieve bottlenecks, but it can also increase mixing, accelerate representation collapse, and increase computational cost. Conversely, removing edges can reduce unnecessary mixing and can even improve information flow when redundancy leads to counterproductive diffusion patterns. This tension motivates approaches that explicitly modify topology while controlling structural drift and complexity. For example, inductive rewiring methods aim to preserve aspects of global structure while remaining usable beyond transductive settings (Arnaiz-Rodríguez et al., 2022). More recently, pruning guided by spectral objectives has been proposed to jointly mitigate over-squashing and oversmoothing (Jamadandi et al., 2024). Empirically and theoretically, the interaction between these phenomena has been studied as a trade-off. More precisely, alleviating one failure mode can amplify the other if topology changes are not controlled (Giraldo et al., 2023). Related analyses also emphasize that depth limitations, node degree effects, and long-range propagation constraints interact in practice (Qureshi, 2023), and that vanishing gradients can couple to both oversmoothing and over-squashing in deep message passing (Arroyo et al., 2025). These results collectively suggest that successful rewiring should not only densify graphs, but should actively control how connectivity is redistributed to relieve bottlenecks without amplifying collapse dynamics.

**Contributions.** In this work, we investigate a simple, resistance-driven rewiring mechanism designed to probe and mitigate over-squashing while controlling densification through an explicit edge budget. At each step, we add an edge between the pair of nodes with the worst (largest) effective resistance to directly strengthen the weakest long-range connectivity, and we simultaneously remove an edge attaining one of the best (smallest) resistance values to limit densification and curb excessive mixing in regions that are already strongly coupled under the same global criterion. The procedure is parameter-free beyond the rewiring budget and relies on a single global measure that aggregates all paths between node pairs. We evaluate the resulting topology changes on two homophilic citation graphs (Cora, CiteSeer) and two heterophilic graphs (Cornell, Texas). We study robustness across common message passing architectures (GCN (Kipf & Welling, 2017)) as well as a directed setting using DirGCN (Tong et al., 2020), for directed graphs, we rely on a directed generalization of effective resistance that is well-defined under appropriate connectivity conditions (Young et al., 2015). Beyond proposing rewiring as an intervention, our core contribution is a representation-level analysis that compares the similarity structure of the initial node representations and the learned embeddings before and after rewiring. This approach lets us distinguish when improvements stem from genuinely better long-range communication versus when they arise from changes in embedding geometry. In turn, this leads to novel conclusions about where over-squashing originates, how targeted structural edits reshape similarity patterns, and when resistance-guided rewiring improves transmissivity rather than merely increasing feature/embedding alignment.

## 2 METHODS

We study node classification on both homophilic graphs (Cora, CiteSeer) and heterophilic graphs (Texas, Cornell), since the alignment between labels and edges strongly affects how message passing behaves.

We consider message passing GNNs where each layer corresponds to one hop of aggregation. For undirected graphs, we use the standard Graph Convolutional Network (GCN) update:

$$\mathbf{H}^{(l+1)} = \sigma\left(\tilde{\mathbf{D}}^{-\frac{1}{2}}\tilde{\mathbf{A}}\tilde{\mathbf{D}}^{-\frac{1}{2}}\mathbf{H}^{(l)}\mathbf{W}^{(l)}\right), \tag{1}$$

where $\tilde{\mathbf{A}} = \mathbf{A} + \mathbf{I}$ adds self-loops, $\tilde{\mathbf{D}}$ is the corresponding degree matrix, $\mathbf{W}^{(l)}$ are trainable weights, and $\sigma(\cdot)$ is a non-linear activation.

When the input graph is directed, symmetrizing $\mathbf{A}$ discards directionality. We therefore adopt a directed GCN variant that separates incoming and outgoing aggregation:

$$\mathbf{H}^{(l+1)} = \sigma\Big(\mathbf{D}_{\text{in}}^{-1}\mathbf{A}_{\text{in}}\mathbf{H}^{(l)}\mathbf{W}_{\text{in}}^{(l)} + \mathbf{D}_{\text{out}}^{-1}\mathbf{A}_{\text{out}}\mathbf{H}^{(l)}\mathbf{W}_{\text{out}}^{(l)} + \mathbf{H}^{(l)}\mathbf{W}_{\text{self}}^{(l)}\Big), \tag{2}$$

where $\mathbf{A}_{\text{in}}$ is the incoming adjacency matrix with $(\mathbf{A}_{\text{in}})_{ij} = 1$ iff $j \to i$, and $\mathbf{A}_{\text{out}}$ is the outgoing adjacency matrix with $(\mathbf{A}_{\text{out}})_{ij} = 1$ iff $i \to j$. The corresponding degree matrices are diagonal: $(\mathbf{D}_{\text{in}})_{ii} = \sum_j (\mathbf{A}_{\text{in}})_{ij}$ and $(\mathbf{D}_{\text{out}})_{ii} = \sum_j (\mathbf{A}_{\text{out}})_{ij}$.

All hyperparameters used are available in Appendix C.1.

## 2.1 Quantifying oversmoothing and over-squashing

Deep message passing exhibits two coupled pathologies. Oversmoothing refers to loss of representational diversity, where embeddings become increasingly similar across layers. Over-squashing refers to attenuation of long-range signals when information must traverse structural bottlenecks and be compressed into fixed-size vectors.

**Oversmoothing metric.** To monitor the evolution of representational collapse, we measure how the similarity between node embeddings changes as depth increases. Oversmoothing manifests as a progressive loss of diversity, where node representations become increasingly aligned and less discriminative across layers. In this work, we focus on cosine similarity as our primary diagnostic, as it directly captures the angular alignment between embeddings and provides an intuitive view of how message passing drives representations toward or away from collapse, see Appendix B.

**Over-squashing metrics.** Over-squashing is primarily a structural phenomenon where long-range information becomes ineffective when it must traverse narrow bottlenecks and be compressed into fixed-size representations. To quantify this effect, we focus on effective resistance as our main topology-level diagnostic, since it captures global, multi-path connectivity between node pairs.

**Effective resistance.** For an undirected connected graph with Laplacian $\mathbf{L}$, the resistance distance between nodes $i$ and $j$ is

$$R_{ij} = (\mathbf{e}_i - \mathbf{e}_j)^\top \mathbf{L}^\dagger (\mathbf{e}_i - \mathbf{e}_j), \tag{3}$$

where $\mathbf{L}^\dagger$ is the Moore-Penrose pseudoinverse of $\mathbf{L}$ and $\mathbf{e}_i$ is the $i$-th standard basis vector. Resistance distance can be interpreted through the electrical-network analogy: it measures how easily "flow" can pass between two nodes when it is allowed to traverse all available paths. Large $R_{ij}$ indicates weak multi-path connectivity, which is consistent with a bottlenecked route for information transmission.

For directed graphs, we use the directed effective resistance (Young et al., 2015). Let $\mathbf{L} = \mathbf{D} - \mathbf{A}$ be the (out-)Laplacian, with $\mathbf{D} = \text{diag}(d^{\text{out}})$, and let $\Pi = \mathbf{I} - \frac{1}{N}\mathbf{1}\mathbf{1}^\top$. Choose any $\mathbf{Q} \in \mathbb{R}^{(N-1)\times N}$ with orthonormal rows spanning $\mathbf{1}^\perp$ (so $\mathbf{Q}^\top\mathbf{Q} = \Pi$), and set $\bar{\mathbf{L}} = \mathbf{Q}\mathbf{L}\mathbf{Q}^\top$. Let $\Sigma$ be the unique solution of the Lyapunov equation

$$\bar{\mathbf{L}}\Sigma + \Sigma\bar{\mathbf{L}}^\top = \mathbf{I}_{N-1}. \tag{4}$$

Define $\mathbf{X} = 2\mathbf{Q}^\top\Sigma\mathbf{Q}$ and the directed effective resistance

$$R_{ij} = (\mathbf{e}_i - \mathbf{e}_j)^\top\mathbf{X}(\mathbf{e}_i - \mathbf{e}_j). \tag{5}$$

In directed experiments, we only evaluate $R_{ij}$ for node pairs that lie in the same strongly connected component (SCC), and ignore pairs in different SCCs.

**Resistance per hop.** To emphasize bottlenecks beyond mere graph distance, we also consider the normalized quantity

$$R_{ij}^{\text{hop}} = \frac{R_{ij}}{d(i,j)}, \tag{6}$$

where $d(i, j)$ is the shortest-path distance. Large $R_{ij}^{\text{hop}}$ highlights pairs that are poorly connected relative to their hop distance.

Beyond its electrical-network interpretation, effective resistance is also theoretically linked to message passing sensitivity. In particular, recent analyses connect pairwise resistance to upper bounds on the influence of node $j$ on node $i$ through the network, by bounding appropriate norms of the Jacobian of the embeddings with respect to the input features (Black et al., 2023). Under these bounds, large $R_{ij}$ implies a stronger attenuation of long-range influence (i.e., a tighter bottleneck), which motivates using resistance as a proxy for over-squashing.

**Curvature.** As a baseline comparison, we report Ollivier-Ricci curvature on edges (Hamilton, 1988):

$$\kappa(u, v) = 1 - \frac{W_1(\mu_u, \mu_v)}{d(u, v)}, \tag{7}$$

where $W_1(\cdot, \cdot)$ is the Wasserstein-1 distance between neighborhood measures $\mu_u$ and $\mu_v$ and $d(u, v)$ is the shortest-path distance. Negative curvature ($\kappa(u, v) < 0$) indicates that the neighborhoods around $u$ and $v$ are, in a transport sense, farther apart than the nodes themselves, which is consistent with an edge acting as a local bridge between poorly overlapping neighborhoods (Topping et al., 2022). Such edges are commonly interpreted as locally bottlenecked connections, and have been used to guide curvature-based rewiring. Here, we treat curvature as a local baseline indicator to contrast with resistance-based, explicitly global diagnostics.

## 2.2 Resistance-based add-remove rewiring

Let $G^{(0)} = (\mathcal{V}, \mathcal{E}^{(0)})$ be the input graph and let $B \in \mathbb{N}$ be the rewiring budget (number of iterations). We construct a sequence $\{G^{(t)}\}_{t=0}^{B}$, $G^{(t)} = (\mathcal{V}, \mathcal{E}^{(t)})$, by performing one add-remove step per iteration.

For undirected graphs, we compute effective resistance for all pairs. For directed graphs, we restrict all pairwise computations to nodes within the same strongly connected component (SCC), since resistances across different SCCs are not finite under the directed definition.

Define the admissible pair set

$$\Omega^{(t)} = \begin{cases} \{(i, j) \in \mathcal{V} \times \mathcal{V} : i \neq j\}, & \text{undirected,} \\ \{(i, j) \in \mathcal{V} \times \mathcal{V} : i \neq j, \ \text{SCC}(i) = \text{SCC}(j)\}, & \text{directed,} \end{cases}$$

and let $R_{ij}^{(t)}$ denote the (undirected or directed) effective resistance computed on $G^{(t)}$.

**Edge addition.** We first identify the maximally resistant pair

$$(u^\star, v^\star) \in \arg \max_{(i,j) \in \Omega^{(t)}} R_{ij}^{(t)}. \tag{8}$$

There are two cases:

1. If $(u^\star, v^\star) \notin \mathcal{E}^{(t)}$, we add the edge $(u^\star, v^\star)$.

2. If $(u^\star, v^\star) \in \mathcal{E}^{(t)}$, we instead add two edges that connect $u^\star$ and $v^\star$ to each other's neighborhoods. Let $\mathcal{N}^{(t)}(\cdot)$ denote the (undirected) neighborhood, for directed graphs, we use the union of in- and out-neighbors. Choose $n_u \in \mathcal{N}^{(t)}(u^\star)$ and $n_v \in \mathcal{N}^{(t)}(v^\star)$ such that the candidate edges do not already exist, then add

$$(u^\star, n_v) \quad \text{and} \quad (v^\star, n_u) \tag{9}$$

(undirected: add $\{u^\star, n_v\}$ and $\{v^\star, n_u\}$). When multiple choices are possible, we apply a deterministic tie-breaking rule which minimize the resistance value.

**Edge removal.** We identify the minimally resistant edge in the current graph,

$$(p^\star, q^\star) \in \arg \min_{(i,j) \in \mathcal{E}^{(t)}} R_{ij}^{(t)}. \tag{10}$$

We remove this edge only if it preserves connectivity. In the directed case, we apply the same principle within SCCs. We remove $(p^\star, q^\star)$ only if it does not split its SCC into multiple SCCs. If the minimizer is not removable, we consider the next-lowest-resistance edge until we find a removable one.

Writing $\mathcal{E}_{\text{add}}^{(t)}$ for the edge(s) added in the addition step, the update is

$$\mathcal{E}^{(t+1)} = \left(\mathcal{E}^{(t)} \cup \mathcal{E}_{\text{add}}^{(t)}\right) \setminus \{(p^\star, q^\star)\}. \tag{11}$$

Overall, this procedure strengthens the weakest pairwise connectivity (large resistance) while pruning redundant connections between already well-coupled nodes (small resistance), subject to preserving global connectivity (undirected) or SCC structure (directed).

**PairNorm.** To mitigate oversmoothing during training, we optionally apply PairNorm (Zhao & Akoglu, 2019) to intermediate embeddings via normalization:

$$\mathbf{H}' = \frac{\mathbf{H} - \bar{\mathbf{H}}}{\sqrt{\frac{1}{|\mathcal{V}|^2} \sum_{i,j} \|\mathbf{h}_i - \mathbf{h}_j\|^2}}, \tag{12}$$

where $\bar{\mathbf{H}}$ is the mean embedding across nodes. This normalization controls the scale of pairwise distances and helps prevent representation collapse across layers.

The algorithmic pipeline pseudocode followed in the project is shared in Appendix A.

## 3 RESULTS

We evaluate whether resistance-driven rewiring improves message passing by targeting oversquashing rather than oversmoothing. We therefore focus on analyzing (i) performance across depth, (ii) implications to an explicit oversmoothing regularizer (PairNorm), and (iii) representation-level diagnostics.

To control the amount of structural modification, we parametrize each rewiring operator by a budget $r$, which specifies an upper bound on the number of edge edits the operator may perform. We evaluate $r \in \{0.01, 0.05, 0.1, 0.15\}$. The operators are not forced to add or remove an edge at every step, an edit is applied only when it is necessary and valid under the operator constraints. As a consequence, the realized number of edits can differ from the nominal budget and can vary across rewiring types, see Appendix C.2.

To characterize how performance evolves with depth, we train models with an increasing number of layers and report test accuracy as a function of depth. We repeat this depth sweep across rewiring strategies and budgets, and include training on the original (non-rewired) graph as a baseline reference.

### 3.1 MITIGATING OVERSMOOTHING

Rewiring changes graph connectivity and can affect depth behavior, we therefore repeat the depth sweep with and without PairNorm to compare settings without oversmoothing regularization to settings with an explicit oversmoothing regularizer.

Figures 1, 3 report GCN test accuracy as a function of depth for several rewiring strategies across four datasets. On the homophilic citation graphs (Cora and CiteSeer), accuracy tends to degrade as depth increases when PairNorm is not used, reflecting the onset of oversmoothing in deeper models. Introducing PairNorm stabilizes performance across layers, preventing the sharp accuracy drop observed in the unregularized setting. In this regime, resistance-based rewiring strategies further help maintain accuracy as depth increases, suggesting that improving global connectivity can complement oversmoothing control by facilitating long-range information propagation without inducing early collapse.

In contrast, the heterophilic graphs (Cornell and Texas) exhibit a different behavior. Here, PairNorm has a weaker effect on the depth dynamics, and performance does not deteriorate as sharply with

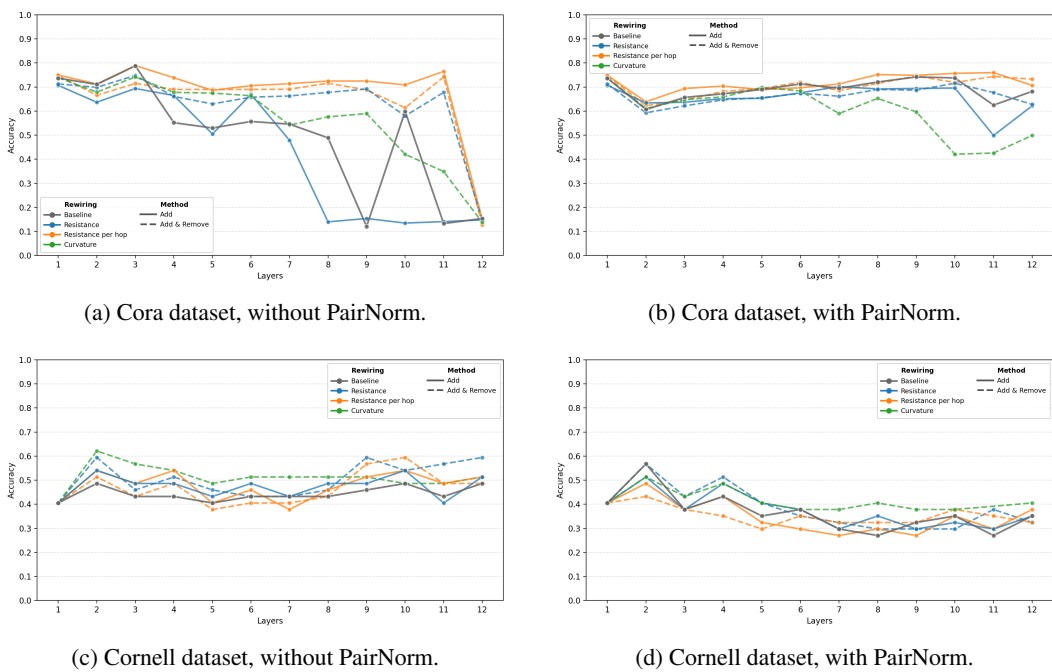

(a) Cora dataset, without PairNorm.

(b) Cora dataset, with PairNorm.

(c) Cornell dataset, without PairNorm.

(d) Cornell dataset, with PairNorm.

Figure 1: GCN test accuracy versus depth (number of layers) under different rewiring strategies.

increasing depth even without explicit normalization. Instead, the dominant factor becomes the graph topology itself as rewiring strategies consistently improve accuracy compared to the baseline across multiple depths.

## 3.2 Representation analysis

Several rewiring strategies achieve similar test accuracy across depths and budgets. We therefore complement the performance curves with an embedding-level analysis, to compare the representations produced by different rewirings and to check whether similar accuracy corresponds to similar or different embeddings.

**Edge-set overlap as a starting point.** We first compare the rewiring operators at the graph level, by analyzing which edges they add or remove under the same budget. The UpSet plots in Figures 6 summarize intersections between the sets of added edges for different rewiring strategies at $r = 0.1$ (Cora, Cornell, CiteSeer and Texas). This establishes the extent to which the methods start from similar or different modified graphs before training, since distinct added-edge sets can lead to different neighborhood structure and message-passing trajectories.

**Linear probe on penultimate-layer embeddings.** We probe the penultimate layer (pre-logits) to assess linear separability of the learned representation independently of the final classifier head, following the standard use of linear probes on intermediate representations (Alain & Bengio, 2016; Tenney et al., 2019). Figure 2 reports linear-probe accuracy across depth for Curvature and Resistance (add and remove) rewiring strategies at budget $r = 0.1$, evaluated both without PairNorm and with PairNorm. On Cora and Cornell, probe accuracy varies with depth in both settings, with differences across rewiring strategies and across the PairNorm/no-PairNorm conditions.

**Cosine similarity between classes.** To study how class-conditional representation similarity evolves through the network, we compute the mean cosine similarity between node embeddings for pairs of nodes in the same class and for pairs in different classes, which provides a representation-level view related to homophily versus heterophily. We report cosine similarity as a function of the inner layer index (with the outer layer fixed), for Cora and Cornell at $r = 0.1$, both without PairNorm and with PairNorm (Figure 7). This diagnostic tracks how within-class similarity and

between-class similarity change across layers under each rewiring strategy, and how these trends differ when PairNorm is applied.

**CKA similarity between representations.** Finally, we compare the learned representations across rewiring strategies using Centered Kernel Alignment (CKA) (Kornblith et al., 2019), which provides a representation-level similarity measure that is invariant to orthogonal transformations and isotropic scaling of features. This is useful for comparing embeddings across independently trained models where neuron bases can rotate or rescale without changing the underlying representation. Figures 4, 5 reports CKA similarity between last-layer representations learned with curvature versus those learned with alternative rewiring strategies across depth, for the four different datasets, without PairNorm and with PairNorm.

This complements the cosine and probe analyses by comparing full representations across methods rather than class-conditioned pairwise similarities or linear separability alone.

All plots about the comparison between both rewiring techniques are available in Appendix C.4.

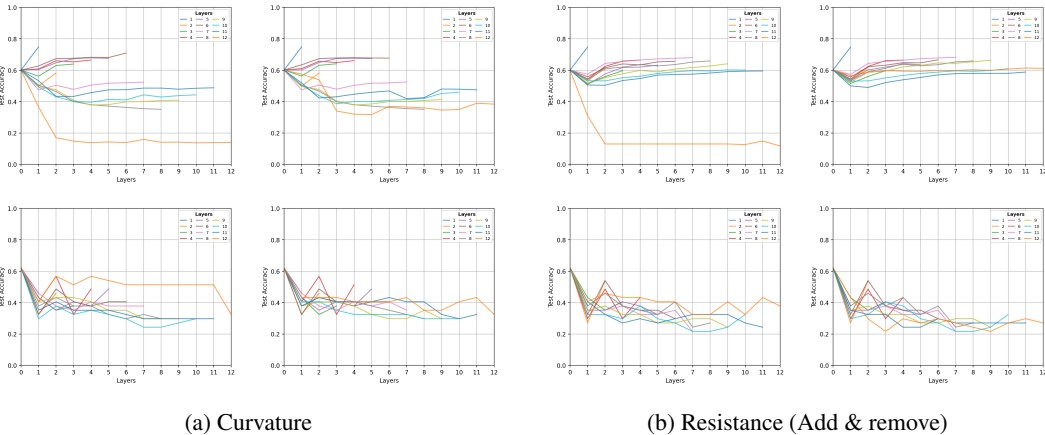

(a) Curvature (b) Resistance (Add & remove)

Figure 2: Linear-probe accuracy on penultimate-layer embeddings of a GCN as depth increases, comparing rewiring strategies at budget $0.1$. Results are shown for Cora (top) and Cornell (bottom), without PairNorm in the left columns and with PairNorm in the right ones; each colored curve corresponds to a different readout (output) layer. PairNorm consistently stabilizes the depth trend of probe accuracy, while resistance-driven rewiring alters the depth at which informative embeddings form and reduces the degradation observed in the unnormalized setting, highlighting the interaction between normalization (oversmoothing control) and topology edits (bottleneck alleviation).

## 4 DISCUSSION

Our experiments highlight a practical trade-off in topology interventions for message passing. We observed that reducing over-squashing by improving global connectivity can simultaneously increase neighborhood mixing and accelerate oversmoothing, and which effect dominates depends on both graph type and network depth.

On homophilic citation graphs, shallow GCNs already perform strongly, and the main degradation with depth is consistent with oversmoothing-driven representation collapse rather than a lack of structural reachability. In this regime, rewiring yields limited gains because additional edges largely act as extra diffusion paths. PairNorm stabilizes deep models and largely removes depth-related collapse, once it is enabled, differences between rewiring strategies become comparatively small. Together, these results suggest that, on homophilic graphs, controlling collapse is the primary lever for depth, while topology correction plays a secondary role.

On heterophilic graphs, rewiring improves performance for shallow and moderately deep models, consistent with the idea that bottlenecks can limit early information flow and that improving weak global connectivity helps distant signals reach target nodes. However, deeper models still degrade even after rewiring. This indicates that reachability alone is not sufficient since repeated aggregation

increasingly mixes signals across classes, and this mixing remains harmful at depth. PairNorm reduces collapse but does not eliminate the depth-induced mixing effect, aligning with the linear-probe behavior on Cornell where separability still degrades with depth. Overall, these results point to depth-induced mixing, not only collapse, as the dominant limitation in heterophilic settings.

Curvature and resistance-based rewiring can reach similar accuracy levels, but they need not produce the same internal solutions. The two theoretical criteria operate at different scales. Curvature emphasizes local geometric bottlenecks, while effective resistance captures global multi-path connectivity. Our representation-level analyses via linear probing and similarity diagnostics suggest that these strategies emphasize different structural aspects of the graph, leading to embeddings that can differ even when end-task accuracy is comparable. This makes the choice of rewiring criterion relevant not only for performance but also for what information the representation preserves.

For directed graphs, restricting resistance computations to pairs within the same strongly connected component keeps the directed resistance well-defined and preserves reachability during rewiring, but it also constrains which bottlenecks can be modified. In practice, this means the directed extension is best viewed as a principled first step toward topology correction under directionality, while acknowledging that the SCC constraint can prevent interventions on certain global obstacles.

Two limitations are central. First, computational cost is nontrivial, especially for directed effective resistance, which makes scaling and frequent recomputation a concern. Second, resistance is task-agnostic, and while lowering resistance improves connectivity in heterophilic graphs it can also create harmful neighbors and amplify class mixing. These observations suggest that topology correction is most reliable when paired with mechanisms that explicitly control mixing, particularly in deep or heterophilic regimes.

### ACKNOWLEDGEMENTS

The author thankfully acknowledges RES resources provided by BSC in MareNostrum to BCV-2025-3-0045.

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

APPENDIX

## A  ALGORITHMIC PIPELINE

---

**Algorithm 1** Algorithmic Pipeline

---

**Require:** Graph $G = (\mathcal{V}, \mathcal{E})$ (directed or undirected), rewiring budget $r$, GNN type $\mathcal{M} \in$ {GCN, DirGCN}, PairNorm flag $\mathcal{P}$
**Ensure:** Trained model parameters and evaluation metrics
 1: **Load data:**
 2: Load graph $G$ in its original directed or undirected form
 3: **Apply rewiring:**
 4: Construct rewired graph $\tilde{G}$ by applying the rewiring operator to $G$ with budget $r$
                                        ▷ Includes edge addition and, if enabled, edge removal
 5: **Train model:**
 6: **if** $\mathcal{M} =$ GCN **then**
 7:     **if** $\tilde{G}$ is directed **then**
 8:         Apply undirecting map to $\tilde{G}$ (e.g., symmetrization)
 9:     **end if**
10:     Train GCN on the resulting undirected graph
11: **else**
12:     Train DirGCN directly on $\tilde{G}$
13: **end if**
14: **Normalization and readout:**
15: **if** $\mathcal{P}$ is enabled **then**
16:     Apply PairNorm at each GNN layer
17: **end if**
18: Compute task loss and evaluation metrics from final node representations

---

## B  OVERSMOOTHING METRIC

In order to measure oversmoothing evolution across the embeddings, we use cosine similarity, as introduced in 2.1. Given node embeddings $\mathbf{h}_i$ and $\mathbf{h}_j$, we compute:

**Cosine similarity.**

$$\text{CosineSimilarity}(\mathbf{h}_i, \mathbf{h}_j) = |\cos(\mathbf{h}_i, \mathbf{h}_j)| = \left| \frac{\mathbf{h}_i^{\top} \mathbf{h}_j}{\|\mathbf{h}_i\| \, \|\mathbf{h}_j\|} \right| \tag{13}$$

## C  EXPERIMENTAL RESULTS

### C.1  EXPERIMENTAL SETUP

We use the standard benchmark datasets and splits as commonly reported in prior work, as well as for hyperparameters. For Cora and CiteSeer, we follow the experimental setup of Kipf & Welling (2017), using the default transductive split convention associated with that benchmark and the same training hyperparameters (hidden dimension 16, dropout 0.5, learning rate 0.01, weight decay $5 \times 10^{-3}$). For Cornell and Texas, we follow Pei et al. (2020), using their dataset split protocol and hyperparameters (hidden dimension 64, dropout 0.5, learning rate 0.01, weight decay $5 \times 10^{-4}$). We load the datasets and their default split objects through PyTorch Geometric (Fey & Lenssen, 2019), and we keep the training pipeline fixed across all experiments: we optimize cross-entropy on the training nodes using Adam, select the checkpoint with the best validation accuracy, and report the corresponding test accuracy. Table 1 summarizes the adopted hyperparameters.

Table 1: Hyperparameters adopted from prior work (no tuning): Kipf & Welling (2017) for Cora/CiteSeer and Pei et al. (2020) for Cornell/Texas.

| Dataset | Hidden dim $d$ | Dropout | lr | Weight decay |
|---------|----------------|---------|------|--------------|
| Cora | 16 | 0.5 | 0.01 | $5 \times 10^{-3}$ |
| CiteSeer | 16 | 0.5 | 0.01 | $5 \times 10^{-3}$ |
| Cornell | 64 | 0.5 | 0.01 | $5 \times 10^{-4}$ |
| Texas | 64 | 0.5 | 0.01 | $5 \times 10^{-4}$ |

## C.2 REWIRING STATS

Table 2: Directed graphs: edge modifications by budget. Each cell reports **added / removed** edges, and on the next line the corresponding percentages relative to the initial number of edges $E_0$. For `res` and `res_hop`, we report **added / –**.

| Dataset | $N$ | $E_0$ | Budget | Curvature | Resistance (Add & Remove) | Resistance per Hop (Add & Remove) | Resistance | Resistance per Hop |
|---------|-----|-------|--------|-----------|---------------------------|------------------------------------|------------|---------------------|
| cora | 2708 | 10556 | 1% | 105 / 105 (1.0% / 1.0%) | 105 / 105 (1.0% / 1.0%) | 105 / 105 (1.0% / 1.0%) | 105 / – (1.0% / –) | 105 / – (1.0% / –) |
| | | | 5% | 517 / 517 (4.9% / 4.9%) | 527 / 527 (5.0% / 5.0%) | 526 / 526 (5.0% / 5.0%) | 527 / – (5.0% / –) | 526 / – (5.0% / –) |
| | | | 10% | 1007 / 1007 (9.5% / 9.5%) | 1055 / 1055 (10.0% / 10.0%) | 1007 / 1007 (9.5% / 9.5%) | 1055 / – (10.0% / –) | 1007 / – (9.5% / –) |
| | | | 15% | 1489 / 1489 (14.1% / 14.1%) | 1583 / 1583 (15.0% / 15.0%) | 1440 / 1440 (13.6% / 13.6%) | 1583 / – (15.0% / –) | 1440 / – (13.6% / –) |
| citeseer | 3327 | 9104 | 1% | 91 / 91 (1.0% / 1.0%) | 91 / 91 (1.0% / 1.0%) | 91 / 91 (1.0% / 1.0%) | 91 / – (1.0% / –) | 91 / – (1.0% / –) |
| | | | 5% | 451 / 451 (5.0% / 5.0%) | 455 / 455 (5.0% / 5.0%) | 455 / 455 (5.0% / 5.0%) | 455 / – (5.0% / –) | 455 / – (5.0% / –) |
| | | | 10% | 895 / 895 (9.8% / 9.8%) | 910 / 910 (10.0% / 10.0%) | 909 / 909 (10.0% / 10.0%) | 910 / – (10.0% / –) | 909 / – (10.0% / –) |
| | | | 15% | 1336 / 1336 (14.7% / 14.7%) | 1365 / 1365 (15.0% / 15.0%) | 1312 / 1312 (14.4% / 14.4%) | 1365 / – (15.0% / –) | 1312 / – (14.4% / –) |
| texas | 183 | 325 | 1% | 3 / 3 (0.9% / 0.9%) | 3 / 3 (0.9% / 0.9%) | 3 / 3 (0.9% / 0.9%) | 3 / – (0.9% / –) | 3 / – (0.9% / –) |
| | | | 5% | 16 / 16 (4.9% / 4.9%) | 16 / 16 (4.9% / 4.9%) | 15 / 15 (4.6% / 4.6%) | 16 / – (4.9% / –) | 15 / – (4.6% / –) |
| | | | 10% | 10 / 24 (3.1% / 7.4%) | 32 / 32 (9.8% / 9.8%) | 31 / 31 (9.5% / 9.5%) | 32 / – (9.8% / –) | 31 / – (9.5% / –) |
| | | | 15% | 32 / 32 (9.8% / 9.8%) | 46 / 46 (14.2% / 14.2%) | 46 / 46 (14.2% / 14.2%) | 46 / – (14.2% / –) | 46 / – (14.2% / –) |
| cornell | 183 | 298 | 1% | 2 / 2 (0.7% / 0.7%) | 2 / 2 (0.7% / 0.7%) | 2 / 2 (0.7% / 0.7%) | 2 / – (0.7% / –) | 2 / – (0.7% / –) |
| | | | 5% | 14 / 14 (4.7% / 4.7%) | 14 / 14 (4.7% / 4.7%) | 12 / 12 (4.0% / 4.0%) | 14 / – (4.7% / –) | 12 / – (4.0% / –) |
| | | | 10% | 26 / 26 (8.7% / 8.7%) | 29 / 29 (9.7% / 9.7%) | 25 / 25 (8.4% / 8.4%) | 29 / – (9.7% / –) | 25 / – (8.4% / –) |
| | | | 15% | 39 / 39 (13.1% / 13.1%) | 40 / 40 (13.4% / 13.4%) | 39 / 39 (13.1% / 13.1%) | 40 / – (13.4% / –) | 39 / – (13.1% / –) |

Table 3: Undirected graphs: edge modifications by budget. Each cell reports added / removed edges, and on the next line the corresponding percentages relative to the initial number of (undirected) edges $E_0$. For res and res_hop, removals are not performed and are shown as "–".

| Dataset | $N$ | $E_0$ | Budget | Curvature | Resistance (Add & Remove) | Resistance per Hop (Add & Remove) | Resistance | Resistance per Hop |
|---------|-----|-------|--------|-----------|----------------------------|------------------------------------|------------|--------------------|
| cora | 2708 | 5278 | 1% | 52 / 52 (1.0% / 1.0%) | 52 / 52 (1.0% / 1.0%) | 52 / 52 (1.0% / 1.0%) | 52 / – (1.0% / –) | 52 / – (1.0% / –) |
| | | | 5% | 263 / 263 (5.0% / 5.0%) | 263 / 263 (5.0% / 5.0%) | 263 / 263 (5.0% / 5.0%) | 263 / – (5.0% / –) | 263 / – (5.0% / –) |
| | | | 10% | 527 / 527 (10.0% / 10.0%) | 519 / 519 (9.8% / 9.8%) | 522 / 522 (9.9% / 9.9%) | 527 / – (10.0% / –) | 527 / – (10.0% / –) |
| | | | 15% | 789 / 789 (14.9% / 14.9%) | 745 / 745 (14.1% / 14.1%) | 730 / 730 (13.8% / 13.8%) | 791 / – (15.0% / –) | 791 / – (15.0% / –) |
| citeseer | 3327 | 4552 | 1% | 45 / 45 (1.0% / 1.0%) | 45 / 45 (1.0% / 1.0%) | 45 / 45 (1.0% / 1.0%) | 45 / – (1.0% / –) | 45 / – (1.0% / –) |
| | | | 5% | 227 / 227 (5.0% / 5.0%) | 227 / 227 (5.0% / 5.0%) | 227 / 227 (5.0% / 5.0%) | 227 / – (5.0% / –) | 227 / – (5.0% / –) |
| | | | 10% | 455 / 455 (10.0% / 10.0%) | 453 / 453 (10.0% / 10.0%) | 454 / 454 (10.0% / 10.0%) | 455 / – (10.0% / –) | 455 / – (10.0% / –) |
| | | | 15% | 682 / 682 (15.0% / 15.0%) | 648 / 648 (14.2% / 14.2%) | 672 / 672 (14.8% / 14.8%) | 682 / – (15.0% / –) | 682 / – (15.0% / –) |
| texas | 183 | 295 | 1% | 3 / 3 (1.0% / 1.0%) | 2 / 2 (0.7% / 0.7%) | 3 / 3 (1.0% / 1.0%) | 3 / – (1.0% / –) | 3 / – (1.0% / –) |
| | | | 5% | 14 / 14 (4.7% / 4.7%) | 13 / 13 (4.4% / 4.4%) | 14 / 14 (4.7% / 4.7%) | 14 / – (4.7% / –) | 14 / – (4.7% / –) |
| | | | 10% | 29 / 29 (9.8% / 9.8%) | 25 / 25 (8.5% / 8.5%) | 29 / 29 (9.8% / 9.8%) | 30 / – (10.2% / –) | 30 / – (10.2% / –) |
| | | | 15% | 44 / 44 (14.9% / 14.9%) | 36 / 36 (12.2% / 12.2%) | 44 / 44 (14.9% / 14.9%) | 44 / – (14.9% / –) | 44 / – (14.9% / –) |
| cornell | 183 | 280 | 1% | 3 / 3 (1.1% / 1.1%) | 2 / 2 (0.7% / 0.7%) | 3 / 3 (1.1% / 1.1%) | 3 / – (1.1% / –) | 3 / – (1.1% / –) |
| | | | 5% | 14 / 14 (5.0% / 5.0%) | 13 / 13 (4.6% / 4.6%) | 14 / 14 (5.0% / 5.0%) | 14 / – (5.0% / –) | 14 / – (5.0% / –) |
| | | | 10% | 30 / 30 (10.7% / 10.7%) | 24 / 24 (8.6% / 8.6%) | 30 / 30 (10.7% / 10.7%) | 30 / – (10.7% / –) | 30 / – (10.7% / –) |
| | | | 15% | 41 / 41 (14.6% / 14.6%) | 30 / 30 (10.7% / 10.7%) | 42 / 42 (15.0% / 15.0%) | 42 / – (15.0% / –) | 42 / – (15.0% / –) |

## C.3 RESULTS OF ADDITIONAL DATASETS

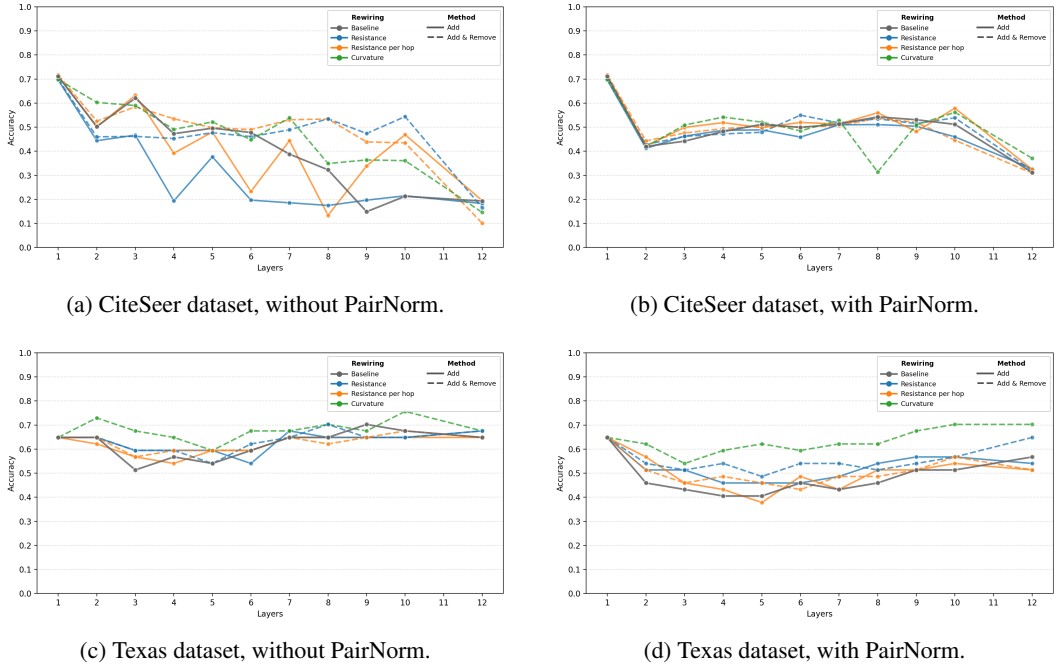

(a) CiteSeer dataset, without PairNorm.

(b) CiteSeer dataset, with PairNorm.

(c) Texas dataset, without PairNorm.

(d) Texas dataset, with PairNorm.

Figure 3: GCN accuracy versus depth (number of layers) under different rewiring strategies.

## C.4 REPRESENTATION-LEVEL DIAGNOSTICS

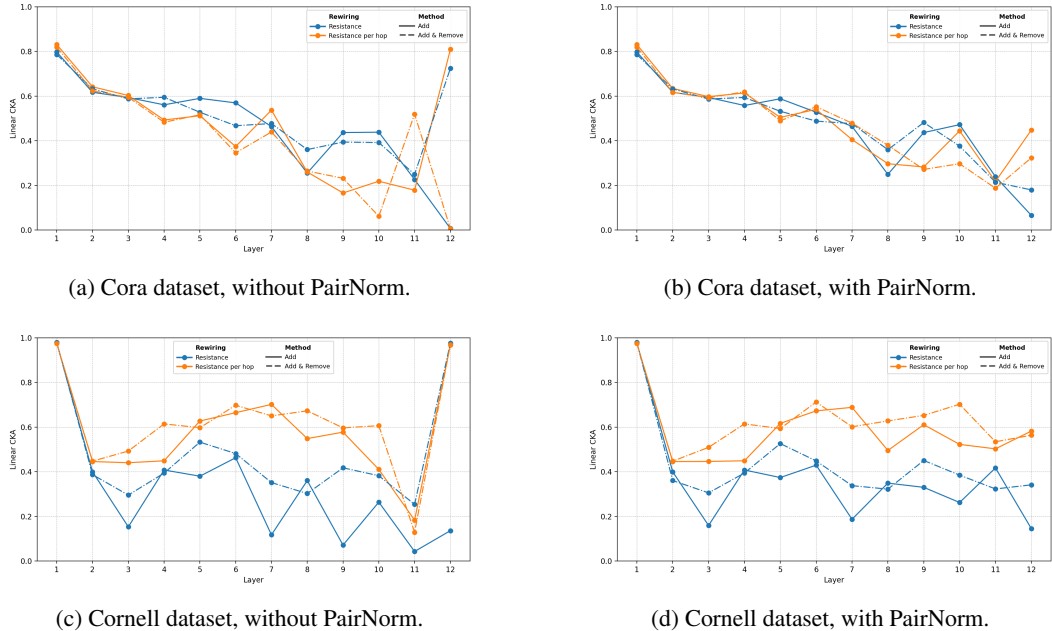

(a) Cora dataset, without PairNorm.

(b) Cora dataset, with PairNorm.

(c) Cornell dataset, without PairNorm.

(d) Cornell dataset, with PairNorm.

Figure 4: CKA similarity between last-layer GCN representations with curvature rewiring (curvature) and proposed rewiring strategies with $0.1$ as budget.

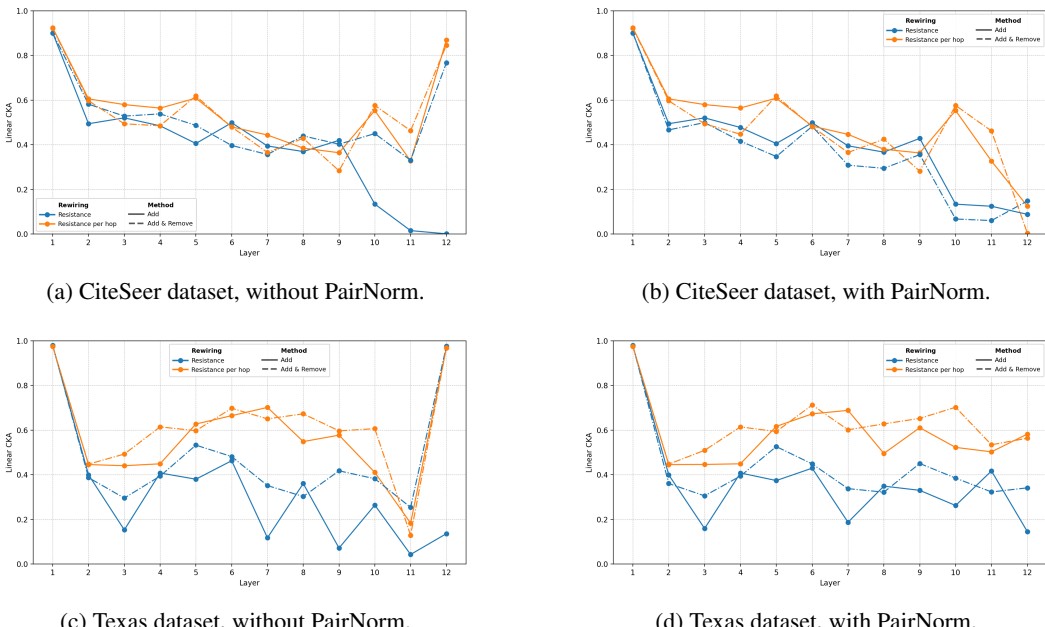

(a) CiteSeer dataset, without PairNorm.

(b) CiteSeer dataset, with PairNorm.

(c) Texas dataset, without PairNorm.

(d) Texas dataset, with PairNorm.

Figure 5: CKA similarity between last-layer GCN representations with curvature rewiring (curvature) and proposed rewiring strategies with $0.1$ as budget.

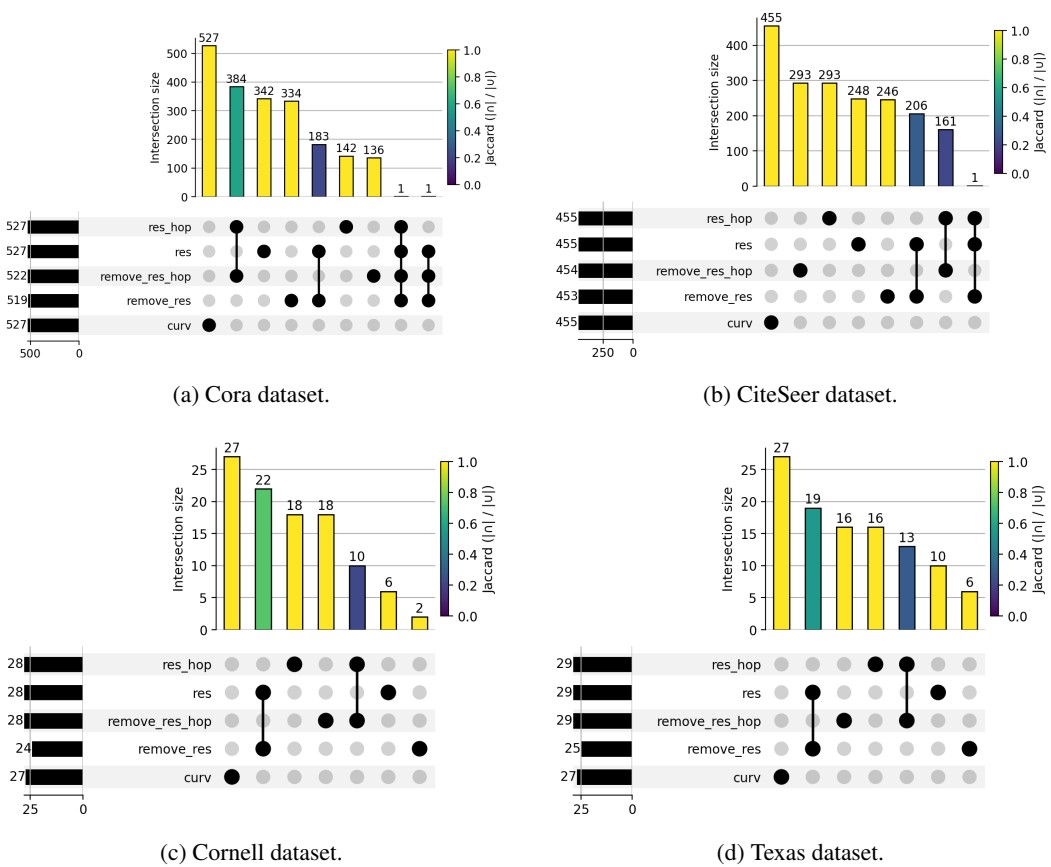

Figure 6: UpSet plots of edges added by different rewiring strategies at budget $r = 0.1$. Bars report intersection sizes of the added-edge sets, bar color indicates Jaccard overlap (intersection over union) for each combination.

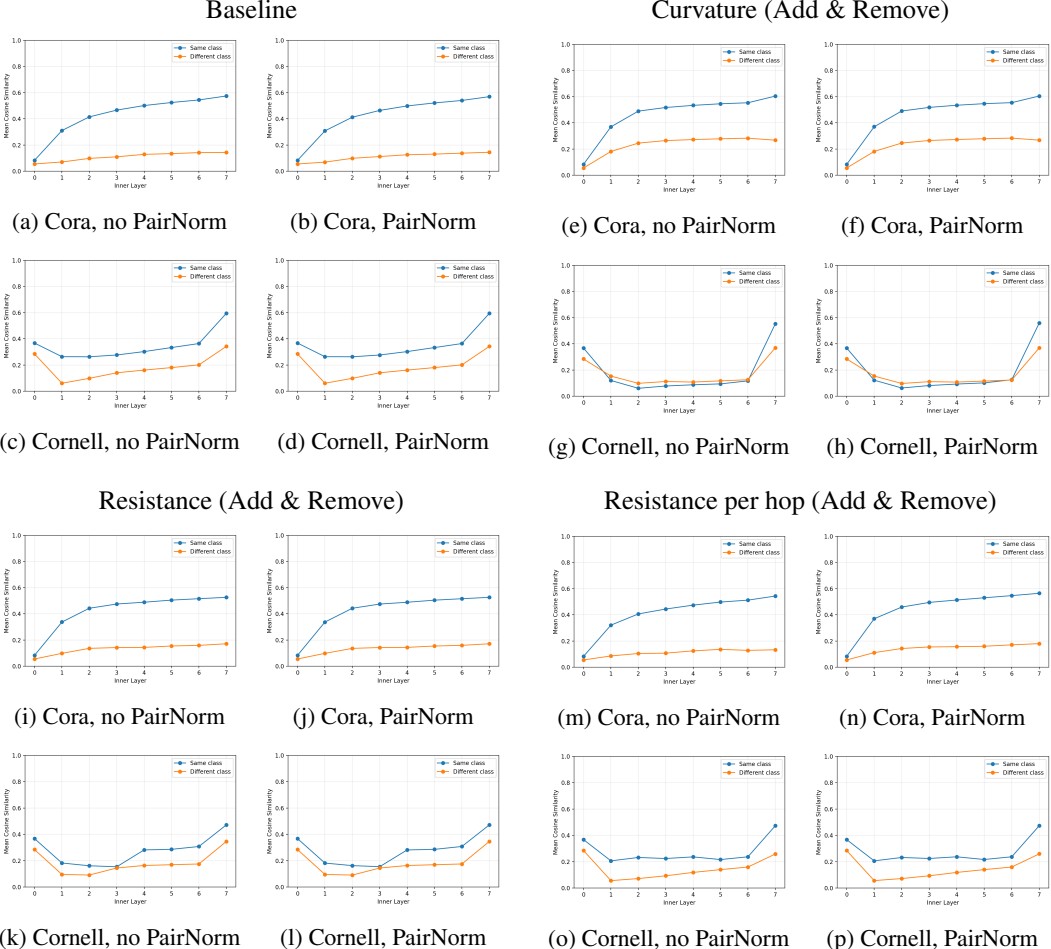

Figure 7: Mean cosine similarity between node embeddings as a function of the inner layer index (outer layer fixed to 7), comparing node pairs from the same class versus different classes, under different rewiring strategies at budget $r = 0.1$. Each grouped panel corresponds to one graph construction: Baseline, Curvature, Resistance, and Resistance per hop. Within each group, results are shown for Cora and Cornell, with and without PairNorm.

## C.5 RESULTS DETAIL

### C.5.1 TABLE RESULTS

Table 4: Directed graphs: best layer / max accuracy at budget 1% across datasets (Cornell, Texas).

| Model | Rewiring | Cornell | Texas |
|---|---|---|---|
| **GCN (PairNorm=No)** | | | |
| | Curvature | 8 / 40.5 | 1 / 64.9 |
| | Resistance (Add & Remove) | 12 / 40.5 | 1 / 64.9 |
| | Resistance per Hop (Add & Remove) | 12 / 40.5 | 1 / 64.9 |
| | Resistance | 9 / 40.5 | 1 / 64.9 |
| | Resistance per Hop | 8 / 40.5 | 1 / 64.9 |
| **GCN (PairNorm=Yes)** | | | |
| | Curvature | 2 / 48.6 | 10 / 70.3 |
| | Resistance (Add & Remove) | 2 / 54.0 | 1 / 64.9 |
| | Resistance per Hop (Add & Remove) | 2 / 54.0 | 12 / 67.6 |
| | Resistance | 2 / 51.4 | 1 / 64.9 |
| | Resistance per Hop | 2 / 51.4 | 1 / 64.9 |
| **DirGCN (PairNorm=No)** | | | |
| | Curvature | 3 / 81.1 | 2 / 81.1 |
| | Resistance (Add & Remove) | 3 / 81.1 | 7 / 83.8 |
| | Resistance per Hop (Add & Remove) | 3 / 81.1 | 4 / 81.1 |
| | Resistance | 3 / 78.4 | 2 / 81.1 |
| | Resistance per Hop | 3 / 83.8 | 2 / 81.1 |
| **DirGCN (PairNorm=Yes)** | | | |
| | Curvature | 3 / 81.1 | 8 / 86.5 |
| | Resistance (Add & Remove) | 5 / 81.1 | 9 / 83.8 |
| | Resistance per Hop (Add & Remove) | 3 / 81.1 | 2 / 81.1 |
| | Resistance | 2 / 78.4 | 2 / 81.1 |
| | Resistance per Hop | 3 / 83.8 | 2 / 81.1 |

Table 5: Directed graphs: best layer / max accuracy at budget 1% across datasets (Cora, Citeseer).

| Model | Rewiring | Cora | Citeseer |
|---|---|---|---|
| **GCN (PairNorm=No)** | | | |
| | Curvature | 3 / 81.0 | 1 / 71.0 |
| | Resistance (Add & Remove) | 3 / 78.4 | 1 / 71.0 |
| | Resistance per Hop (Add & Remove) | 3 / 79.1 | 1 / 70.7 |
| | Resistance | 3 / 77.9 | 1 / 71.0 |
| | Resistance per Hop | 3 / 78.6 | 1 / 71.4 |
| **GCN (PairNorm=Yes)** | | | |
| | Curvature | 3 / 81.0 | 1 / 71.0 |
| | Resistance (Add & Remove) | 3 / 78.4 | 1 / 71.0 |
| | Resistance per Hop (Add & Remove) | 3 / 79.1 | 1 / 70.7 |
| | Resistance | 3 / 77.9 | 1 / 71.0 |
| | Resistance per Hop | 3 / 78.6 | 1 / 71.4 |
| **DirGCN (PairNorm=No)** | | | |
| | Curvature | 4 / 75.8 | 2 / 68.4 |
| | Resistance (Add & Remove) | 2 / 74.7 | 2 / 68.2 |
| | Resistance per Hop (Add & Remove) | 4 / 75.2 | 2 / 68.2 |
| | Resistance | 2 / 74.6 | 2 / 67.9 |
| | Resistance per Hop | 4 / 75.7 | 2 / 68.1 |
| **DirGCN (PairNorm=Yes)** | | | |
| | Curvature | 4 / 76.4 | 2 / 68.4 |
| | Resistance (Add & Remove) | 2 / 76.1 | 2 / 68.2 |
| | Resistance per Hop (Add & Remove) | 5 / 76.5 | 2 / 68.2 |
| | Resistance | 3 / 75.4 | 2 / 67.9 |
| | Resistance per Hop | 4 / 76.5 | 2 / 68.1 |

Table 6: Directed graphs: best layer / max accuracy at budget 5% across datasets (Cornell, Texas).

| Model | Rewiring | Cornell | Texas |
|---|---|---|---|
| | Curvature | 10 / 40.5 | 12 / 67.6 |
| | Resistance (Add & Remove) | 9 / 40.5 | 1 / 64.9 |
| | Resistance per Hop (Add & Remove) | 10 / 40.5 | 1 / 64.9 |
| | Resistance | 12 / 48.6 | 1 / 64.9 |
| | Resistance per Hop | 9 / 40.5 | 1 / 64.9 |
| **GCN (PairNorm=Yes)** | | | |
| | Curvature | 2 / 46.0 | 1 / 64.9 |
| | Resistance (Add & Remove) | 10 / 46.0 | 10 / 67.6 |
| | Resistance per Hop (Add & Remove) | 2 / 46.0 | 1 / 64.9 |
| | Resistance | 2 / 51.4 | 9 / 67.6 |
| | Resistance per Hop | 2 / 51.4 | 1 / 64.9 |
| **DirGCN (PairNorm=No)** | | | |
| | Curvature | 4 / 81.1 | 7 / 83.8 |
| | Resistance (Add & Remove) | 3 / 86.5 | 4 / 83.8 |
| | Resistance per Hop (Add & Remove) | 4 / 86.5 | 8 / 83.8 |
| | Resistance | 3 / 83.8 | 4 / 81.1 |
| | Resistance per Hop | 3 / 89.2 | 2 / 81.1 |
| **DirGCN (PairNorm=Yes)** | | | |
| | Curvature | 2 / 81.1 | 2 / 83.8 |
| | Resistance (Add & Remove) | 3 / 86.5 | 3 / 81.1 |
| | Resistance per Hop (Add & Remove) | 3 / 86.5 | 12 / 83.8 |
| | Resistance | 3 / 86.5 | 2 / 78.4 |
| | Resistance per Hop | 3 / 89.2 | 2 / 81.1 |

Table 8: Directed graphs: best layer / max accuracy at budget 10% across datasets (Cornell, Texas).

| Model | Rewiring | Cornell | Texas |
|---|---|---|---|
| | Curvature | 1 / 40.5 | 1 / 64.9 |
| | Resistance (Add & Remove) | 1 / 40.5 | 2 / 67.6 |
| | Resistance per Hop (Add & Remove) | 1 / 40.5 | 1 / 64.9 |
| | Resistance | 12 / 40.5 | 1 / 64.9 |
| | Resistance per Hop | 10 / 40.5 | 1 / 64.9 |
| **GCN (PairNorm=Yes)** | | | |
| | Curvature | 3 / 43.2 | 1 / 64.9 |
| | Resistance (Add & Remove) | 2 / 43.2 | 2 / 73.0 |
| | Resistance per Hop (Add & Remove) | 8 / 46.0 | 1 / 64.9 |
| | Resistance | 2 / 43.2 | 1 / 64.9 |
| | Resistance per Hop | 2 / 51.4 | 1 / 64.9 |
| **DirGCN (PairNorm=No)** | | | |
| | Curvature | 3 / 83.8 | 7 / 86.5 |
| | Resistance (Add & Remove) | 2 / 75.7 | 3 / 81.1 |
| | Resistance per Hop (Add & Remove) | 3 / 78.4 | 5 / 83.8 |
| | Resistance | 3 / 81.1 | 2 / 75.7 |
| | Resistance per Hop | 3 / 86.5 | 6 / 86.5 |
| **DirGCN (PairNorm=Yes)** | | | |
| | Curvature | 2 / 81.1 | 2 / 81.1 |
| | Resistance (Add & Remove) | 2 / 78.4 | 2 / 75.7 |
| | Resistance per Hop (Add & Remove) | 3 / 78.4 | 4 / 83.8 |
| | Resistance | 3 / 83.8 | 4 / 81.1 |
| | Resistance per Hop | 3 / 81.1 | 2 / 81.1 |

Table 7: Directed graphs: best layer / max accuracy at budget 5% across datasets (Cora, Citeseer).

| Model | Rewiring | Cora | Citeseer |
|---|---|---|---|
| | Curvature | 3 / 75.8 | 1 / 70.3 |
| | Resistance (Add & Remove) | 1 / 71.7 | 1 / 70.1 |
| | Resistance per Hop (Add & Remove) | 1 / 74.2 | 1 / 72.0 |
| | Resistance | 1 / 71.7 | 1 / 70.7 |
| | Resistance per Hop | 3 / 74.2 | 1 / 72.1 |
| **GCN (PairNorm=Yes)** | | | |
| | Curvature | 3 / 75.1 | 1 / 70.3 |
| | Resistance (Add & Remove) | 6 / 74.4 | 1 / 70.1 |
| | Resistance per Hop (Add & Remove) | 2 / 77.0 | 1 / 72.0 |
| | Resistance | 5 / 75.7 | 1 / 70.7 |
| | Resistance per Hop | 6 / 77.6 | 1 / 72.1 |
| **DirGCN (PairNorm=No)** | | | |
| | Curvature | 2 / 72.5 | 1 / 66.1 |
| | Resistance (Add & Remove) | 2 / 74.1 | 2 / 69.0 |
| | Resistance per Hop (Add & Remove) | 2 / 75.0 | 2 / 70.6 |
| | Resistance | 2 / 72.2 | 2 / 69.4 |
| | Resistance per Hop | 2 / 75.8 | 2 / 70.5 |
| **DirGCN (PairNorm=Yes)** | | | |
| | Curvature | 5 / 73.6 | 1 / 66.1 |
| | Resistance (Add & Remove) | 2 / 74.1 | 2 / 68.0 |
| | Resistance per Hop (Add & Remove) | 2 / 76.0 | 2 / 70.5 |
| | Resistance | 2 / 72.9 | 2 / 69.4 |
| | Resistance per Hop | 2 / 77.5 | 2 / 70.5 |

Table 9: Directed graphs: best layer / max accuracy at budget 10% across datasets (Cora, Citeseer).

| Model | Rewiring | Cora | Citeseer |
|---|---|---|---|
| | Curvature | 3 / 73.8 | 1 / 69.8 |
| | Resistance (Add & Remove) | 1 / 69.6 | 1 / 69.3 |
| | Resistance per Hop (Add & Remove) | 1 / 73.3 | 1 / 71.8 |
| | Resistance | 1 / 70.4 | 1 / 68.9 |
| | Resistance per Hop | 3 / 78.7 | 1 / 71.9 |
| **GCN (PairNorm=Yes)** | | | |
| | Curvature | 3 / 73.6 | 1 / 69.8 |
| | Resistance (Add & Remove) | 2 / 73.0 | 1 / 69.3 |
| | Resistance per Hop (Add & Remove) | 3 / 77.6 | 1 / 71.8 |
| | Resistance | 2 / 73.3 | 1 / 68.9 |
| | Resistance per Hop | 3 / 78.7 | 1 / 71.9 |
| **DirGCN (PairNorm=No)** | | | |
| | Curvature | 3 / 71.1 | 1 / 65.7 |
| | Resistance (Add & Remove) | 2 / 73.6 | 2 / 67.5 |
| | Resistance per Hop (Add & Remove) | 4 / 76.8 | 2 / 71.4 |
| | Resistance | 2 / 73.5 | 2 / 68.9 |
| | Resistance per Hop | 2 / 75.7 | 2 / 71.3 |
| **DirGCN (PairNorm=Yes)** | | | |
| | Curvature | 4 / 74.6 | 3 / 66.9 |
| | Resistance (Add & Remove) | 3 / 73.7 | 2 / 67.5 |
| | Resistance per Hop (Add & Remove) | 5 / 78.5 | 2 / 71.4 |
| | Resistance | 2 / 73.4 | 2 / 68.9 |
| | Resistance per Hop | 4 / 78.0 | 2 / 71.3 |

Table 10: Directed graphs: best layer / max accuracy at budget 15% across datasets (Cornell, Texas).

| Model | Rewiring | Cornell | Texas |
|---|---|---|---|
| **GCN (PairNorm=No)** | | | |
| | Curvature | 9 / 46.0 | 1 / 64.9 |
| | Resistance (Add & Remove) | 10 / 40.5 | 2 / 70.3 |
| | Resistance per Hop (Add & Remove) | 1 / 40.5 | 1 / 64.9 |
| | Resistance | 7 / 43.2 | 2 / 67.6 |
| | Resistance per Hop | 1 / 37.8 | 1 / 64.9 |
| **GCN (PairNorm=Yes)** | | | |
| | Curvature | 8 / 46.0 | 1 / 64.9 |
| | Resistance (Add & Remove) | 2 / 40.5 | 2 / 70.3 |
| | Resistance per Hop (Add & Remove) | 9 / 43.2 | 1 / 64.9 |
| | Resistance | 10 / 43.2 | 10 / 67.6 |
| | Resistance per Hop | 2 / 40.5 | 1 / 64.9 |
| **DirGCN (PairNorm=No)** | | | |
| | Curvature | 3 / 83.8 | 5 / 81.1 |
| | Resistance (Add & Remove) | 3 / 83.8 | 4 / 81.1 |
| | Resistance per Hop (Add & Remove) | 3 / 78.4 | 5 / 83.8 |
| | Resistance | 3 / 75.7 | 4 / 83.8 |
| | Resistance per Hop | 4 / 81.1 | 5 / 83.8 |
| **DirGCN (PairNorm=Yes)** | | | |
| | Curvature | 3 / 81.1 | 4 / 83.8 |
| | Resistance (Add & Remove) | 3 / 86.5 | 2 / 78.4 |
| | Resistance per Hop (Add & Remove) | 3 / 78.4 | 3 / 81.1 |
| | Resistance | 3 / 81.1 | 2 / 81.1 |
| | Resistance per Hop | 3 / 81.1 | 2 / 83.8 |

Table 11: Directed graphs: best layer / max accuracy at budget 15% across datasets (Cora, Citeseer).

| Model | Rewiring | Cora | Citeseer |
|---|---|---|---|
| **GCN (PairNorm=No)** | | | |
| | Curvature | 3 / 73.8 | 1 / 69.2 |
| | Resistance (Add & Remove) | 1 / 69.5 | 1 / 67.8 |
| | Resistance per Hop (Add & Remove) | 3 / 74.4 | 1 / 71.7 |
| | Resistance | 1 / 68.3 | 1 / 67.3 |
| | Resistance per Hop | 4 / 76.0 | 1 / 71.3 |
| **GCN (PairNorm=Yes)** | | | |
| | Curvature | 1 / 71.1 | 1 / 69.2 |
| | Resistance (Add & Remove) | 2 / 70.6 | 1 / 67.8 |
| | Resistance per Hop (Add & Remove) | 2 / 77.8 | 1 / 71.7 |
| | Resistance | 2 / 71.8 | 1 / 67.3 |
| | Resistance per Hop | 2 / 76.9 | 1 / 71.3 |
| **DirGCN (PairNorm=No)** | | | |
| | Curvature | 2 / 67.2 | 1 / 64.9 |
| | Resistance (Add & Remove) | 2 / 72.8 | 2 / 68.5 |
| | Resistance per Hop (Add & Remove) | 2 / 75.5 | 2 / 70.4 |
| | Resistance | 3 / 72.6 | 1 / 66.3 |
| | Resistance per Hop | 2 / 77.7 | 1 / 68.9 |
| **DirGCN (PairNorm=Yes)** | | | |
| | Curvature | 3 / 73.0 | 3 / 65.0 |
| | Resistance (Add & Remove) | 3 / 73.0 | 2 / 67.4 |
| | Resistance per Hop (Add & Remove) | 4 / 77.5 | 2 / 70.4 |
| | Resistance | 2 / 72.4 | 1 / 66.3 |
| | Resistance per Hop | 3 / 78.8 | 4 / 69.1 |

Table 12: Undirected graphs: best layer / max accuracy at budget 1% across datasets (Cornell, Texas).

| Model | Rewiring | Cornell | Texas |
|---|---|---|---|
| **GCN (PairNorm=No)** | | | |
| | Curvature | 2 / 48.6 | 9 / 73.0 |
| | Resistance (Add & Remove) | 9 / 59.5 | 12 / 70.3 |
| | Resistance per Hop (Add & Remove) | 10 / 59.5 | 9 / 70.3 |
| | Resistance | 10 / 54.0 | 9 / 67.6 |
| | Resistance per Hop | 9 / 48.6 | 9 / 70.3 |
| **GCN (PairNorm=Yes)** | | | |
| | Curvature | 2 / 54.0 | 10 / 67.6 |
| | Resistance (Add & Remove) | 10 / 59.5 | 9 / 67.6 |
| | Resistance per Hop (Add & Remove) | 9 / 56.8 | 9 / 67.6 |
| | Resistance | 2 / 56.8 | 10 / 70.3 |
| | Resistance per Hop | 11 / 59.5 | 12 / 67.6 |

Table 13: Undirected graphs: best layer / max accuracy at budget 1% across datasets (Cora, Citeseer).

| Model | Rewiring | Cora | Citeseer |
|---|---|---|---|
| | Curvature | 3 / 80.7 | 1 / 70.8 |
| | Resistance (Add & Remove) | 3 / 78.7 | 1 / 70.9 |
| | Resistance per Hop (Add & Remove) | 3 / 79.8 | 1 / 70.8 |
| | Resistance | 3 / 78.4 | 1 / 70.9 |
| | Resistance per Hop | 3 / 79.6 | 1 / 71.5 |
| **GCN (PairNorm=Yes)** | | | |
| | Curvature | 3 / 80.7 | 1 / 70.7 |
| | Resistance (Add & Remove) | 3 / 78.7 | 1 / 70.9 |
| | Resistance per Hop (Add & Remove) | 3 / 79.8 | 1 / 70.8 |
| | Resistance | 3 / 78.4 | 1 / 70.9 |
| | Resistance per Hop | 3 / 79.6 | 1 / 71.5 |

Table 14: Undirected graphs: best layer / max accuracy at budget 5% across datasets (Cornell, Texas).

| Model | Rewiring | Cornell | Texas |
|---|---|---|---|
| **GCN (PairNorm=No)** | | | |
| | Curvature | 2 / 56.8 | 2 / 70.3 |
| | Resistance (Add & Remove) | 12 / 59.5 | 9 / 70.3 |
| | Resistance per Hop (Add & Remove) | 2 / 51.4 | 10 / 67.6 |
| | Resistance | 2 / 54.0 | 12 / 70.3 |
| | Resistance per Hop | 2 / 54.0 | 9 / 67.6 |
| **GCN (PairNorm=Yes)** | | | |
| | Curvature | 2 / 54.0 | 2 / 70.3 |
| | Resistance (Add & Remove) | 2 / 56.8 | 12 / 67.6 |
| | Resistance per Hop (Add & Remove) | 11 / 54.0 | 12 / 67.6 |
| | Resistance | 2 / 59.5 | 8 / 70.3 |
| | Resistance per Hop | 9 / 59.5 | 12 / 70.3 |

Table 15: Undirected graphs: best layer / max accuracy at budget 5% across datasets (Cora, Citeseer).

| Model | Rewiring | Cora | Citeseer |
|---|---|---|---|
| **GCN (PairNorm=No)** | | | |
| | Curvature | 1 / 71.9 | 1 / 70.6 |
| | Resistance (Add & Remove) | 1 / 73.0 | 1 / 70.1 |
| | Resistance per Hop (Add & Remove) | 1 / 74.5 | 1 / 71.2 |
| | Resistance | 1 / 72.2 | 1 / 69.8 |
| | Resistance per Hop | 3 / 79.7 | 1 / 70.7 |
| **GCN (PairNorm=Yes)** | | | |
| | Curvature | 3 / 77.1 | 1 / 70.5 |
| | Resistance (Add & Remove) | 3 / 77.4 | 1 / 70.1 |
| | Resistance per Hop (Add & Remove) | 4 / 77.1 | 1 / 71.2 |
| | Resistance | 2 / 76.4 | 1 / 69.8 |
| | Resistance per Hop | 3 / 79.7 | 1 / 70.7 |

Table 16: Undirected graphs: best layer / max accuracy at budget 10% across datasets (Cornell, Texas).

| Model | Rewiring | Cornell | Texas |
|---|---|---|---|
| **GCN (PairNorm=No)** | | | |
| | Curvature | 2 / 56.8 | 10 / 75.7 |
| | Resistance (Add & Remove) | 2 / 56.8 | 8 / 70.3 |
| | Resistance per Hop (Add & Remove) | 2 / 48.6 | 10 / 67.6 |
| | Resistance | 12 / 51.4 | 7 / 67.6 |
| | Resistance per Hop | 2 / 54.0 | 1 / 64.9 |
| **GCN (PairNorm=Yes)** | | | |
| | Curvature | 2 / 56.8 | 12 / 70.3 |
| | Resistance (Add & Remove) | 2 / 54.0 | 10 / 67.6 |
| | Resistance per Hop (Add & Remove) | 2 / 51.4 | 1 / 64.9 |
| | Resistance | 2 / 51.4 | 7 / 73.0 |
| | Resistance per Hop | 2 / 54.0 | 1 / 64.9 |

Table 17: Undirected graphs: best layer / max accuracy at budget 10% across datasets (Cora, Citeseer).

| Model | Rewiring | Cora | Citeseer |
|---|---|---|---|
| **GCN (PairNorm=No)** | | | |
| | Curvature | 3 / 74.2 | 1 / 69.8 |
| | Resistance (Add & Remove) | 3 / 74.8 | 1 / 70.0 |
| | Resistance per Hop (Add & Remove) | 1 / 74.8 | 1 / 71.7 |
| | Resistance | 1 / 70.7 | 1 / 69.6 |
| | Resistance per Hop | 3 / 78.9 | 1 / 71.3 |
| **GCN (PairNorm=Yes)** | | | |
| | Curvature | 1 / 73.9 | 1 / 69.9 |
| | Resistance (Add & Remove) | 3 / 74.3 | 1 / 70.0 |
| | Resistance per Hop (Add & Remove) | 2 / 77.9 | 1 / 71.7 |
| | Resistance | 2 / 73.9 | 1 / 69.6 |
| | Resistance per Hop | 3 / 78.8 | 1 / 71.3 |

Table 18: Undirected graphs: best layer / max accuracy at budget 15% across datasets (Cornell, Texas).

| Model | Rewiring | Cornell | Texas |
|---|---|---|---|
| **GCN (PairNorm=No)** | | | |
| | Curvature | 2 / 62.2 | 2 / 73.0 |
| | Resistance (Add & Remove) | 2 / 56.8 | 8 / 70.3 |
| | Resistance per Hop (Add & Remove) | 2 / 43.2 | 1 / 64.9 |
| | Resistance | 2 / 54.0 | 8 / 67.6 |
| | Resistance per Hop | 10 / 54.0 | 8 / 67.6 |
| **GCN (PairNorm=Yes)** | | | |
| | Curvature | 2 / 62.2 | 8 / 73.0 |
| | Resistance (Add & Remove) | 2 / 54.0 | 12 / 73.0 |
| | Resistance per Hop (Add & Remove) | 9 / 43.2 | 1 / 64.9 |
| | Resistance | 2 / 56.8 | 8 / 70.3 |
| | Resistance per Hop | 2 / 54.0 | 12 / 67.6 |

Table 19: Undirected graphs: best layer / max accuracy at budget 15% across datasets (Cora, Citeseer).

| Model | Rewiring | Cora | Citeseer |
|---|---|---|---|
| **GCN (PairNorm=No)** | | | |
| | Curvature | 1 / 73.6 | 1 / 69.8 |
| | Resistance (Add & Remove) | 1 / 69.3 | 1 / 69.1 |
| | Resistance per Hop (Add & Remove) | 1 / 75.1 | 1 / 72.5 |
| | Resistance | 1 / 69.9 | 1 / 68.5 |
| | Resistance per Hop | 3 / 74.9 | 1 / 70.8 |
| **GCN (PairNorm=Yes)** | | | |
| | Curvature | 3 / 73.7 | 1 / 69.8 |
| | Resistance (Add & Remove) | 2 / 72.8 | 1 / 69.1 |
| | Resistance per Hop (Add & Remove) | 2 / 76.9 | 1 / 72.5 |
| | Resistance | 2 / 72.4 | 1 / 68.5 |
| | Resistance per Hop | 3 / 78.6 | 1 / 70.8 |

