# OpenReview forum: "Effective Resistance Rewiring: A Simple Topological Correction for Over-Squashing"
_ICLR.cc/2026/Workshop/GRaM — ICLR 2026 Workshop GRaM Poster_

### Official Review · Reviewer_LsLU · 2026-02-12
**Good representation-level analysis but needs stronger experimental validation**

**Rating:** 5
**Confidence:** 3

**Review:**

**Summary:** This paper proposes Effective Resistance Rewiring (ERR), a graph topology modification strategy that adds edges between node pairs with the highest effective resistance and removes edges with the lowest effective resistance, under a fixed edge budget. The method is extended to directed graphs and mitigates oversquashing, but to avoid oversmoothing, the authors apply PairNorm to intermediate embeddings.

**Strengths**
1. The paper is well-suited to GRaM since it presents geometric and spectral perspectives on graph structure.
2. The extension to directed graphs via directed effective resistance is a useful practical contribution.
3. I appreciate the analysis beyond accuracy numbers by using linear probes, CKA, cosine similarity, UpSet plots.
4. The discussion section acknowledges when rewiring helps and when it doesn't, and the oversmoothing trade-off is well articulated.

**Weaknesses**
1. Limited novelty: the method combines well-known components (effective resistance from Black et al., curvature rewiring from Topping et al., and PairNorm from Zhao & Akoglu).
2. Experimental evaluation lacks rigor: no variance reporting and Cornell is a tiny dataset with known instability. Moreover, other datasets could be chosen to discuss long-range/oversquashing.
3. Figure quality is poor and significantly hampers readability.
4. The paper does not consistently demonstrate that ERR outperforms the curvature baseline, undermining the motivation for a resistance-based approach.

**Overall:** The representation-level analysis is the main contribution of this work, since the resistance-driven rewiring mechanism does not present significant novelty and ERR variants sometimes is similar to the curvature baseline (this might be a consequence of the high variance usually reported on datasets like Cornell).


**Suggestions/questions:**
1. I couldn't understand Table 2. Can you provide an interpretation of the behavior of the different colors/readouts?
2. Have you considered evaluating on long-range graph benchmarks (e.g., LRGB) where over-squashing is more clearly the dominant bottleneck? Given the computational cost, perhaps on graph classification datasets (Mutag, Proteins, Enzymes) such as in Black et al? I would feel much more confident about your conclusions if you achieved similar results for such datasets.
3. How many random train/val/test splits were used, and are the reported results averaged over multiple splits?

**Pmlr Suitability:**

Yes

---

### Official Review · Reviewer_RByz · 2026-02-20
**Strong Global Motivation, but Limited Empirical and Conceptual Novelty**

**Rating:** 4
**Confidence:** 3

**Review:**

This paper proposes Effective Resistance Rewiring (ERR), a topology modification method aimed at mitigating over-squashing in message-passing GNNs. The approach uses effective resistance as a global bottleneck diagnostic, adding edges between high-resistance node pairs while removing low-resistance edges to maintain a fixed edge budget. The framework is extended to directed graphs using a Lyapunov-based definition of directed effective resistance. Experiments are conducted on homophilic (Cora, CiteSeer) and heterophilic (Cornell, Texas) benchmarks, with additional analysis of depth behavior, oversmoothing interactions, and representation-level diagnostics.

This work presents a conceptually clean and theoretically grounded rewiring mechanism based on effective resistance:

$$
R_{ij} = (e_i - e_j)^\top L^\dagger (e_i - e_j),
$$

which captures global multi-path connectivity between node pairs. The add–remove strategy is intuitive: strengthen the weakest (highest-resistance) pairs while pruning redundant (lowest-resistance) edges, preserving connectivity or SCC structure in directed graphs.

The global nature of resistance is a meaningful contrast to curvature-based rewiring, which is more local. The directed extension via the Lyapunov equation formulation is technically sound and broadens applicability beyond undirected settings. The paper is clearly written, and the algorithm is straightforward to understand and implement (at least conceptually).

The experimental section is careful and includes depth sweeps, linear probe analysis, cosine similarity across classes, and CKA similarity between models. The authors make a genuine effort to disentangle over-squashing from oversmoothing and to analyze interactions with PairNorm. The observation that alleviating bottlenecks can accelerate mixing, especially in heterophilic graphs, is an important and well-articulated takeaway.

However, the overall novelty is moderate. The connection between effective resistance and over-squashing has been previously explored, and resistance has already been theoretically tied to Jacobian sensitivity and long-range signal attenuation. The primary contribution here is the specific add–remove rewiring mechanism and its empirical study, rather than a fundamentally new theoretical insight.

The empirical improvements are present but not consistently strong. On homophilic datasets, gains are small and largely washed out when PairNorm is applied. On heterophilic datasets, improvements appear at shallow or moderate depth, but deeper models still degrade. The representation analyses are interesting, but they do not clearly demonstrate that resistance-based rewiring provides a decisive advantage over curvature-based alternatives.

A significant practical limitation is computational cost. All-pairs effective resistance requires computing a Laplacian pseudoinverse (or solving a Lyapunov equation in the directed case), which scales poorly. The paper acknowledges this but does not provide scalability analysis or approximation strategies. This weakens the practical impact of the method.

In summary, this is a well-motivated and carefully analyzed workshop paper with solid technical grounding. The conceptual framing is strong, but the novelty and empirical impact are somewhat limited. The work fits well within a geometry/topology-focused workshop context, though it would require stronger scalability or theoretical advances for main-track significance.

**Pmlr Suitability:**

Yes

---

### Official Review · Reviewer_qDdE · 2026-02-21
**Nice enough idea but would benefit from additional/more suitable experiments, and experiment presentation and discussion needs improvement**

**Rating:** 5
**Confidence:** 4

**Review:**

**Summary:**
- The authors present a static graph rewiring method based on effective resistance; a global connectivity complement to existing curvature-based rewiring methods, as a means of permitting long-range interactions and reducing over-squashing
- The authors evaluate several iterations (both purely edge-additive and additive-and-subtractive) on two standard homophilic benchmarks and two standard heterophilic benchmarks, for GCN (and a directed version), and compare against curvature-based rewiring
- The authors include some additional representation-level analysis to study overlap between rewiring methods
- The authors conclude that resistance-based rewiring can mitigate over-squashing by improving global connectivity, but its benefits depend on depth and require normalisation to avoid exacerbating over-smoothing.

**Strengths:**
- The paper concerns GNNs, and Section 1 argues for a geometric perspective based on resistance/random walks; I would therefore consider the area of this paper suitable for inclusion for the GRaM workshop
- The paper is reasonably well-written, the introduction in particular is well-motivated and clear
- I am not aware of a previously proposed resistance-distance based rewiring method, so the method is novel
- I am glad to see the embedding-level analysis; i.e. a level of analysis of the method beyond mere downstream performance

**Weaknesses:**
- The space of rewiring-based OSM/OSQ mitigation strategies is already well developed, I would suggest the authors articulate more clearly what qualitatively new capability ERR enables beyond existing approaches.
- The authors talk about long-range at length but consider only four very basic homophilic and heterophilic datasets, and only one GNN, a GCN. The authors evaluate across GNN depths, but this is less informative for basic short-range benchmarks. The authors should discuss the plethora of research discussing long-range interactions, several long-range benchmarks. At the bare minimum this should be discussed, and some attempt at experiments involving LR should be included, if LR is a central area of this paper [1,2,3,4,5,6]
	- On this note, the authors should be careful of conflating both heterophily (lines 407+) and the over-squashing phenomenon (176) with long-range interactions [7,8]
	- Example, line 400: "On homophilic citation graphs, shallow GCNs already perform strongly... in this regime, rewiring yields limited gains" — so why not expand your analysis to other regimes?
- The figures are atrocious and need fixing to be up to PMLR standard.
	- Content-wise:
		- It should be obvious what message the reader is supposed to get from each figure. These figures contain a lot of subfigures with a lot of very noisy lines that are poorly labeled. They should be condensed to only the most important information, and clear (in the main text, at least)
			- This is particularly troublesome for the representation analysis. You use four different diagnostics, include a lot of plots, most of which are in the appendix and not explained or discussed at any great length. I think the authors would be better off focusing on one or two especially informative diagnostics and investing in explaining clearly *what they show*.
		- Experiments should include error bars and be over multiple seeds
		- Methods shown in figures (e.g. 'remove_res', 'remove_res_hop', etc) are not clearly defined in paper but are left for the reader to figure out
	- Style-wise:
		- Heavily pixelated; use pdf/scalable graphics
		- Tiny legends and axis labels
		- Per the style guide, do not include figure titles
		- Use proper subfigures
		- Suggestion: methods should be colour- or marker-coded so individual methods' results are clear at a glance
- The paper ought to include a more thorough discussion of existing rewiring approaches [9,10,11,12,13]. The authors benchmark their methods only against curvature-based rewiring, this should be expanded
- The authors do not discuss computational cost beyond saying it is 'nontrivial'; more detail please (appendix is fine)
- No code, non-reproducible experiments

**Nits**:
- 079: "curvature criteria are often local in nature, while random-walk quantities such as commute time can be informative but less direct as a design signal for topology interventions."
	- Can you elaborate on this?
- 135: "this leads to novel conclusions about where over-squashing originates, how targeted structural edits reshape similarity patterns, and when resistance-guided rewiring improves transmissivity rather than merely increasing feature/embedding alignment."
	- This feels a little vague and inflated, could you be more specific?
- 171: "We therefore evaluate embedding dispersion using three complementary metrics, each comparing node representations from a different perspective"
	- Can you elaborate on why they are complementary? What is the benefit of these different perspectives?
- 127: "We study robustness across common message passing architectures" — only GCN (and its directed equivalent) is studied

---

[1] https://arxiv.org/abs/2206.08164

[2] https://arxiv.org/abs/2309.00367

[3] https://openreview.net/forum?id=2jf5x5XoYk

[4] https://arxiv.org/abs/2512.17762

[5] https://arxiv.org/abs/2506.05971

[6] https://arxiv.org/abs/2503.09008

[7] https://arxiv.org/abs/2511.20406

[8] https://arxiv.org/abs/2505.15547v3

[9] https://arxiv.org/abs/1911.05485

[10] https://arxiv.org/abs/2305.08018

[11] https://arxiv.org/abs/2310.01668

[12] https://arxiv.org/abs/2210.11790

[13] https://arxiv.org/abs/1907.10903

**Pmlr Suitability:**

Yes

---

### Official Review · Reviewer_Swd3 · 2026-02-23
**Well-motivated but empirically limited**

**Rating:** 5
**Confidence:** 3

**Review:**

## Summary:
This paper proposes Effective Resistance Rewiring (ERR), a spectral method to mitigate over-squashing in message-passing GNNs. It uses effective resistance as graph distance to identify connectivity bottlenecks and rewire the graph under a fixed edge budget. The method is extended to directed graphs using effective resistance and preserves strongly connected components.

## Strengths:
1. This paper studies the topological and spectral aspects of GNNs, which make it suitable for Gram.
2. Using resistance as a global complement to curvature-based rewiring is a reasonable and well-motivated choice.
3. The extension of effective resistance to directed graphs and the preservation of strongly connected components is practically meaningful, as many real-world relational graphs are directed.

## Weaknesses:
1. Results are reported without variance across multiple seeds, especially Cornell is a small dataset with high instability.
2. While the add-remove formulation is new, the connection between effective resistance and over-squashing has already been explored in prior work.
3. The figures are crowded and difficult to interpret, with many subplots and noisy curves.

## Overall
This paper is well-motivated and strongly aligned with the geometric perspectivee of Gram. However, the empirical validation is limited and the novelty is incremental relative to existing work. Strengthening the experimental scope (e.g. different seeds, computational cost) would improve this work.

**Pmlr Suitability:**

Yes

---

### Meta-Review · Area_Chair_T9dY · 2026-02-23

**Decision:**

Accept

**Metareview:**

The authors introduce Effective Resistance Rewiring (ERR) to alleviate over squashing in MPNNs. The reviewers generally appreciated the paper's contributions and found it relevant to the venue. I strongly recommend that the authors incorporate the reviewers' comments into the final version of their paper.

**Relevance To Proceedings:**

Yes — suitable for PMLR (long paper)

**Relevance To Workshop:**

Yes — suitable for GRaM

---

### Decision · Program_Chairs · 2026-03-02

Accept (Poster)